# PRIPRO—Privacy Profiles: User Profiling Management for Smart Environments

**Jonas Cesconetto [1]**, **Luís Augusto Silva [2]**, **Fabricio Bortoluzzi [1]**, **María Navarro-Cáceres [2]**, **Cesar A. Zeferino [1]** and and **Valderi R. Q. Leithardt [1,3,4,5,*]**

[1] Laboratory of Embedded and Distributed Systems, University of Vale do Itajai, Itajaí 88302-901, Brazil; jonascesconetto@edu.univali.br (J.C.); fb@univali.br (F.B.); zeferino@univali.br (C.A.Z.)

[2] Expert Systems and Applications Lab, Faculty of Science, University of Salamanca, Plaza de los Caídos s/n, 37008 Salamanca, Spain; luisaugustos@usal.es (L.A.S.); maria90@usal.es (M.N.-C.)

[3] COPELABS, Universidade Lusófona de Humanidades e Tecnologias, 1749-024 Lisboa, Portugal

[4] Departamento de Informática, Universidade da Beira Interior, 6201-601 Covilhã, Portugal

[5] VALORIZA, Research Center for Endogenous Resources Valorization, Instituto Politécnico de Portalegre, 7300-555 Portalegre, Portugal

**\*** Correspondence: valderi@ipportalegre.pt

**Abstract:** Smart environments are pervasive computing systems that provide higher comfort levels on daily routines throughout interactions among smart sensors and embedded computers. The lack of privacy within these interactions can lead to the exposure of sensitive data. We present PRIPRO (PRIvacy PROfiles), a management tool that includes an Android application that acts on the user's smartphone by allowing or blocking resources according to the context, in order to address this issue. Back-end web server processes and imposes a protocol according to the conditions that the user selected beforehand. The experimental results show that the proposed solution successfully communicates with the Android Device Administration framework, and the device appropriately reacts to the expected set of permissions imposed according to the user's profile with low response time and resource usage.

**Keywords:** data privacy; taxonomy; internet of things; smart environments; user profile management

## 1. Introduction

Many contemporary activities rely on the use of smartphones and other smart mobile devices (SMDs)—e.g., tablets, smartwatches—to promote a better quality of life [1]. Besides, the technological advances enable adding more sensors to these devices and intelligence to our day-to-day environments, which became smart environments. However, these smart environments require access to users' data to achieve the aforementioned intelligence, and they must state how the user's data will be used and ask for following users' privacy preferences [2]. Furthermore, the environment must take care of the information on the users' profiles.

Several works have presented proposals for identifying users from their devices [1], creating social communities based on the user's context, profile and social networks [3], authentication of users with guaranteed privacy and access control in smart environments [4–7], and the detection of groups and activities in smart buildings [8], among others. In this context, in [9], the author presented the middleware UbiPri (Ubiquitous Privacy) for the control and management of data privacy based on a generic model. This middleware comprises several modules that can operate separately. One of these modules is the focus of this paper.

We have conducted several studies to ensure privacy in smart environments based on the UbiPri middleware. In [10], we developed a solution to dynamically manage user profiles according to each

environment's characteristics. This solution is integrated into the PRIPRO (PRIvacy PROfiles) module of UbiPri middleware. This module is responsible for managing user profiles in smart environments and it makes them safer by delimiting resources and services for each profile. In that solution, the profile progression is performed based on the user's frequency in the environment. In [11], we compared classification algorithms to identify the most viable alternative for implementation in the UbiPri notification manager's decision engine. In [12], we presented a proposal for software modeling for the implementation of PRIPRO on the Android operating system (OS). That work also presented the environmental requirements and definitions to carry out user profiles' evolution and apply restrictions/permissions of the resources available on the user's mobile device. These works sought to provide privacy to smart environments that integrate systems, applications, and other technologies directed to the Internet of Things (IoT).

Following the direction of the works that are summarized above, this work aims to develop the PRIPRO module, while taking into account the definitions that are presented in the proposed taxonomy to manage the user's device. The proposed solution seeks to ensure that user profiles can automatically progress and regress in a smart environment to ensure security and privacy for users and the environment. The solution implemented runs on Android and imposes the privacy premises established in the module. We summarize the main contributions:

- We propose a taxonomy to establish metrics, parameters, and hierarchy criteria for the evolution of users' profiles in smart environments.
- We present the deployment and assessment of a solution for the evolution of the user profile in smart environments that are based on data privacy.

The remainder of this article is organized into six sections. Section 2 presents the background and Section 3 discusses related work. Next, Section 4 describes the taxonomy proposed for the evolution of the user's profiles and Section 5 presents the architecture of the proposed solution. Following, Section 6 describes the test scenarios used in the experiments and discusses the obtained results. In conclusion, Section 7 gives the final remarks and discusses future work.

## 2. Background

### 2.1. Smart Environments

We can see a smart environment as a small interconnected world where sensors and devices work collaboratively. The term smart is due to its ability to obtain and apply knowledge autonomously and adapt to the needs of its occupants to improve the experience in the environments [13]. A smart environment aims to facilitate human life in several ways, such as controlling various devices and providing some services [14]. For that, smart environments are equipped with sensors for collecting information and actuators for providing services.

Recent studies, such as those that are presented in [8,15], have shown that the relationship between users and environments is an essential aspect to provide appropriate adaptations and context to the applications. Daily activities vary individually (e.g., walking, typing) and in groups (e.g., meetings, classes) in a stochastic manner. As a result, it is necessary to recognize the different levels of human activities.

Smart environments also use wireless sensor networks (WSN), which often feed the system with external data in addition to the data from the environment sensors. According to [16], a WSN is a collection of low-cost, small-size sensor nodes for sensing certain conditions or events. An example of a WSN-based solution is presented in [17]. The authors propose a privacy-preserving data gathering scheme that explores homomorphic data to keep confidentiality in sensory readings by preventing traffic analysis and flow tracking in wireless sensor networks. The system employs homomorphic encryption functions to make it difficult for attackers to recover data read from the sensors.

## 2.2. Ubiquitous Computing

The exchange of information between devices within a smart environment is essential for the correct functioning of the environment. When dealing with ubiquitous systems, the exchange of information is part of the entire ecosystem. Without data gathering, there is no way to process information without requiring human actions. Some problems arose when the information exchanged was sensitive. For information to be exchanged efficiently, compatibility between devices must be guaranteed while using a Gateway or Web services [10].

Ubiquitous computing is already present in our daily lives. In this context, the inter connectivity of objects results in benefits in the performance of trivial actions, relying on the integration of the physical and digital worlds, and connecting people and objects through sensors. These devices meet the demands of users and collect data from those users, as we can see on smartphones, smart TVs, and smartwatches. The most important technologies are those that disappear from the eyes and they are incorporated into the routine, automating our day, as Weiser stated in 1999 [18].

## 2.3. Smartphones, Privacy And Security

Smartphones are the most used SMD [19], and according to [20], the number of smartphone users worldwide equals 3.5 billion in 2020, and must reach 3.9 billion users by the end of 2021. These devices are equipped with processors with high computational power, large amounts of memory, highly functional operating systems, and built-in sensors. These sensors produce events that can be exploited to recognize user activities and contexts automatically. Such features enable the development of more responsive and adaptable mobile applications that are capable of providing social and environmental awareness to the user.

A smartphone is protected by a permission-based mechanism that limits third-party applications from accessing sensitive resources, such as the internal file system, microphone, Bluetooth, message database, and many others. The problem with the current permission structure is that once an application gains access to a specific resource, it remains privileged for indeterminate time and circumstances. Several applications tend to request more permissions than those that would usually be considered to be appropriate. They do not enforce the dynamic exchange of permissions based on any defined context, and a static way of handling permissions is not appropriate in these cases. The associated security risks depend very much on the current state. Whereas, obtaining a specific resource is harmless in one context, it may violate users' privacy in another context [21].

On the other hand, enterprise environments often require administrative control over employees' devices, for conformity with security policies. In this sense, smartphone operating systems provide resources for the development of security-aware applications. For instance, the Android OS offers the Android Device Administration API (Application Program Interface), which enables a developer to create solutions for information technology (IT) professionals that required control over employee devices [22]. Such device administration resources enable a system administrator to implement security policies for remote and local devices. These policies can be hardcoded into the application, or the application can dynamically fetch the policies from a third-party server. If the user does not obey the policies, it is up to the application to decide how to deal with the situation, usually preventing the user from synchronizing new data.

## 2.4. Indoor Positioning

Many applications require the position of users to provide services. However, identifying the user's exact location in a closed environment is a challenge to be overcome. The Global Navigation Satellite System (GNSS) alone cannot effectively provide user positioning in such environments, and other technologies need to be explored. For example, 5G technology offers greater accuracy and, thus, increases the reliability and availability of positioning information compared to current technologies [23]. The Ultra Wide Band (UWB) technology is also an ally to the internal positioning

systems, being a low energy consumption solution that offers precision in the centimeter, with a data transfer rate of 27 Mbps and a range from 70 m to 250 m [24]. Additionally, UWB can be incorporated into 5G antennas in order to improve range for location applications [25].

## 2.5. Privacy Policy

Some applications inform their security and privacy conduct in the form of statements, contained in privacy policies, in order to protect data and comply with different laws. Within the privacy policy, access control is who defines which subjects can access which objects and in what way. The user determines which applications and who can access their information [14].

Privacy policies act as access management and enable users to define restrictions on object access. For example, children should not access the configuration of smart objects that are dangerous for them or compromise their safety at home (e.g., the access doors of the residence) [26].

The Android operating system, for example, limits access to sensitive resources through the permissions mechanism. Permission is a sequence of specialized content that can be defined by the system or by the developers. There are currently more than 140 permissions defined in the Android application framework, as indicated in the Android's documentation. These permissions range from Internet access to disabling full mobile functions, such as the camera and other sensors [21].

Permissions may be required when an application is cooperating with device features, including calling API functions and read and write operations on the internal file system. Application developers need to incorporate these functionalities into the mobile device's manifest file to use the mobile device's protected functionalities.

A possible solution that does not depend on the operating system used is presented in [27]. In that work, the authors present the design and implementation of a fog-based attack detection framework for IoT. The framework proposed satisfies IoT's distinct requirements, such as distribution, resource limitations, and lower detection time, and it is based on an Extreme Learning Machine (ELM) Semi-Supervised Fuzzy C-Means (ESFCM) method in order to provide efficient attack detection in IoT.

## 2.6. Privacy Profiles

With the availability of several sensors that are integrated into mobile devices, applications now carry sensitive personal information that allows for them to identify, classify, or generate a device user profile [28]. Thus, several difficulties that are related to the treatment and management of privacy must be taken into account, since the data generated in different environments will also be related to devices and users [10].

Changes in the environment promote changes in the context. Any adaptation requires information from sensors, which can be temperature values, fire alarm, among others. Acquiring context information from users, aligning the location, and combining them with the user's profile ensures a symbiosis for smart environments [29].

User profiling is a unique set of hierarchical information that is generated by a smart environment about someone who is interacting with it. The environment needs to access users' personal information to particularize policies, model hierarchies, and define each profile. However, users have the power to allow or disallow access to their data. Without access to the necessary information, applications that use data from this context do not perform their functions effectively.

## 2.7. The UbiPri Middleware

The narrowing of the time-to-market for the development of computer systems requires solutions that help programmers in the design of applications and services, abstracting programming, the heterogeneity of networks, operating systems, and programming languages [30]. The middleware is inserted in this context and it consists of a software layer that provides a uniform computational model and enables remote object invocation, notifications and database access, and distributed data processing.

Usually, a middleware is user-centered. However, in [9], the author proposed a middleware, named UbiPri (Ubiquitous Privacy), for the control and management of privacy with attributions focused on the environment in which the user is. In other words, UbiPri places the environment as the centerpiece of the system, and the other elements must follow the rules and criteria that are determined by the environment.

The main difference between UbiPri and other existing middleware is the definition of a generic model for privacy control and management. UbiPri is service-oriented. It receives data from sensors that are present in the environment, allowing new services to be added to the environment. As we can see in Figure 1, the middleware comprises a set of modules, each one of them with specific characteristics, including:

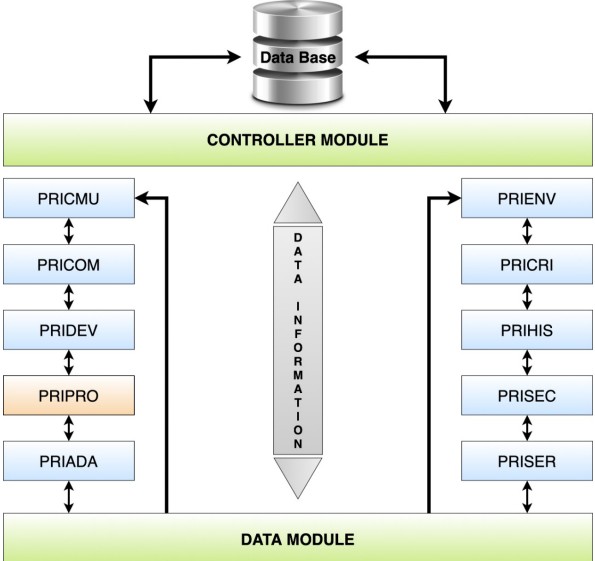

**Figure 1.** Architecture of the UbiPri middleware. The PRIPRO (PRIvacy PROfiles) module controls transactions related to user profile management [9].

- Data Base: contains all of the information necessary for the control and management of the privacy mechanism in ubiquitous environments.
- Controller module: has the purpose of receiving access requests and carrying out the control of the database directly in the database tables.
- Data module: responsible for processing all variables and parameters from other modules. It also has the function of receiving various data and treating them, generating a single output of information for each processing performed.
- PRICMU: deals with the definitions of characteristics related to the user's individual preferences, such as temperature, brightness, and authorized shares (information that the user wants to share with other users and with the environment).
- PRICOM: deals with the forms of communication and how these forms are used within the ubiquitous environment.
- PRIDEV: treats data about devices that can be from the environment itself or that can interact with it.
- PRIPRO: performs control transactions that are related to user profile management.
- PRIADA: performs the management and control of adaptation and is responsible for handling information related to the adaptation of software and hardware in the ubiquitous environment.
- PRIENV: makes the registration of attributes that are related to the environment. This information enables the checking and management what makes up the environment,

- PRICRI: has the rules and definitions of criteria for the environment, such as access, use, sharing, location, and other variables, which can be included, changed, modified, or replaced according to the environment's definitions.
- PRIHIS: stores and handles information related to the user's history, environment, devices, and other variables that can be added later to obtain contextual information.
- PRISEC: performs security-related controls and management, both for the user and the environment.
- PRISER: does the service management of the environment. It handles information about the availability of services that are used individually in each environment.

All of the modules work independently, with characteristics that vary according to the rules previously established in the environment. Therefore, for a module to be able to operate correctly, access to the data contained in the database is allowed.

In this work, we present a solution for implementing the PRIPRO module that is based on a taxonomy that assists in defining the set of rules for the smart environment. As shown below, the proposed solution stands out in comparison to related works that are described in the literature.

## 3. Related Work

In this section, we analyze related work and present their main characteristics to position the proposed approach concerning the current scenario of solutions for user privacy in smart environments.

In [1], the authors presented a study on the identification of users from the data profiles that are generated by their devices. The authors collected information about when, where, and how users use their networked devices to obtain time and event profiles. The time profile represents when a user communicates or transfers data, and the event profile contains the user's frequency at a given location. The authors considered three different events in order to analyze the results: voice calls, text messages, and data transfer, including the combination of two or more types of events. The results demonstrated the feasibility of identifying the user based on the profiles considered. The experiments also showed that the accuracy of this identification increases with the period considered for analysis and the number of data points used. As a work limitation, the experiments that were carried out were restricted to telephone devices, not considering typical IoT devices.

The authors of [3] presented the Spontaneous Social Network (SSN) model that allows for users to interact anywhere and at any time without any pre-existing relationship. The SSN model seeks to create social communities considering a combination of multiple contexts, such as profile, location, and data obtained from external social networks. The SSN model focuses on suggesting social networks that users are interested in. The authors proposed an ontology to determine the user's main context dimensions and implemented the SSN Grouping Mechanism algorithm that calculates the similarity between instances, grouping them based on their degree of similarity. After forming the groups, an application layer provides services and content from the virtual community, promoting social interactions. The experimental results showed a high rate of precision, which indicated that the clustering algorithm could recommend many interesting topics for users. However, it was observed the need for determining an ideal limit for the clustering mechanism. It was also noticed that the more similar the people who make up the domain, the easier it is to form groups.

In the study presented in [4], the author proposed an authentication and reputation management that guarantees data privacy to users through the use of pseudonyms that prevent the exposure of the users' personal information. The author utilizes an ontology to recognize the risks that are associated with authorization and recognition of users. This ontology relies on the estimated degree of confidence and the criticality of the data resources to be accessed. The solution is hosted in the cloud and manages requests and authorizations. Three services are responsible for storing security policies and maintaining and ensuring the reliability of users/entities. The work's objective is to introduce a dynamic process in the decision when granting or denying request access to a user. The authorization is based on the applicant's degree of confidence and the criticality of the data.

In [5], the authors address problems that arise from user awareness and the plug-and-play nature of home IoT environments. The proposed solution relies on attribute-based access control (ABAC) [31] and uses NIST (National Institute of Standards and Technology, Gaithersburg, MD, USA) NGAC (Next Generation Access Control) architecture as a reference. ABAC is the next generation of access control models for obtaining context-aware, refined, and logical and access control in complex heterogeneous systems. Following this approach, the authors categorize the entities in the home IoT environment as subject and object, and then list the various possible attributes. These attributes are obtained from the device manufacturer's specifications and by testing the device in the laboratory. They then describe the different policy categories appropriate to the environment. These categories are designed to be generic and adapt to most home IoT networks. With that, the authors explore the solution in the face of several IoT scenarios. Several domain challenges were put to the test with the developed environment, while using general guidelines that effectively integrate NIST NGAC to any network and environment.

The study that is presented in [8] addresses the problem of understanding occupant activities in a building for providing occupants with comfort and an intelligent indoor environment through building management systems. The authors propose a framework for automated group activity detection and recognition (GADAR) while using beacons associated with the environment and users' smartphones. The system achieves more than 89% accuracy in detecting groups and recognizing group activities. The presented solution has some challenges that are related to the variation of group size and individual actions, the detection of members and group interactions, and issues concerning the privacy and scalability of users.

The research discussed in [6] describes a logical approach and a systematic analysis that presents the main aspects of a new Access Control (AC) model for IoT environments called Pervasive-Based Access Control (PerBAC). The work presents a quantitative and qualitative evaluation and a comparison of the proposed approach with two widely used access control models. The results show some useful aspects in constructing a robust model of access control for IoT. The purpose of the AC process is to limit the operations that each authorized user can perform. In this way, the AC aims to prevent the execution of suspicious actions that could disturb the security of the system.

In [7], the authors present a model for privacy-preserving access control and a framework for secure service provision and composition. The proposed solution is based on the ABAC model and it is able to dynamically verify the compliance between the service providers' privacy policies and the users' privacy preferences. The model also features a series of composite services classified according to the user's preferences and the level of sensitivity to their personal data. The security policies of the system are automatically established.

The works described above show the relevance of issues that are related to identifying the environment, users, and context, mainly the use of solutions to provide data privacy and handle changes in the user profile. In [8], the authors use the context in which the user is to determine the best time for the application to act. When considering access technologies in the environment, the author of [4] introduces a dynamic process in the decision to grant access to users in the environment. The solution that is presented in [1] stands out for using communication between IoT devices to identify user profiles. In this article, we propose an approach that differs from the others by managing the privacy profile in smart environments with a system that allows for the evolution of the user profile in different scenarios.

The works that are described above show the relevance of issues related to identifying the environment, users, and context, mainly the use of solutions to provide data privacy and handle changes in the user profile. In [8], the authors use the context in which the user is to determine the best time for the application to act. When considering access technologies in the environment, the author of [4] introduces a dynamic process in the decision to grant access to users in the environment. The solution that is presented in [1] stands out for using communication between IoT devices to identify user profiles. In this article, we propose an approach that differs from the others by managing

the privacy profile in smart environments with a system that allows the evolution of the user profile in different scenarios. Table 1 summarizes the analyzed works and positions our proposal concerning the most relevant characteristics of these works.

**Table 1.** Related work.

| Work | Year | Environment Identification | User Identification | Context Identification | Data Privacy | User Profile Evolution |
|------|------|:--------------------------:|:-------------------:|:----------------------:|:------------:|:----------------------:|
| [1] | 2017 | ● | ● | | ● | |
| [3] | 2018 | | ● | ● | | |
| [4] | 2018 | | ● | ● | ● | |
| [5] | 2018 | ● | ● | ● | ● | |
| [8] | 2019 | ● | | ● | | |
| [6] | 2019 | ● | ● | ● | ● | |
| [7] | 2019 | | ● | | ● | |
| This work | 2020 | ● | ● | ● | ● | ● |

## 4. A Taxonomy for User's Profile Management

The elements are defined by a taxonomy structure and label the data. Thus, they enable the systematic organization of the relevant data in the form of a hierarchy. The keywords and concepts used to define a taxonomy establish parameters throughout the information production cycle, in which distributed professionals can participate in the knowledge creation process in an organized manner [14,32].

Table 2 presents the works used to compose the requirements of the taxonomic model. Based on these requirements, we defined a set of rules and components necessary to control and manage users' privacy profiles in a smart environment. Figure 2 presents the proposed taxonomy, which is discussed below.

**Table 2.** Works used to define the taxonomic model.

| Component | Works |
|-----------|-------|
| Communication | [14,33–36] |
| Environment | [26,37–43], |
| User | [44–48] |
| Privacy | [2,9,19,21,49–53] |

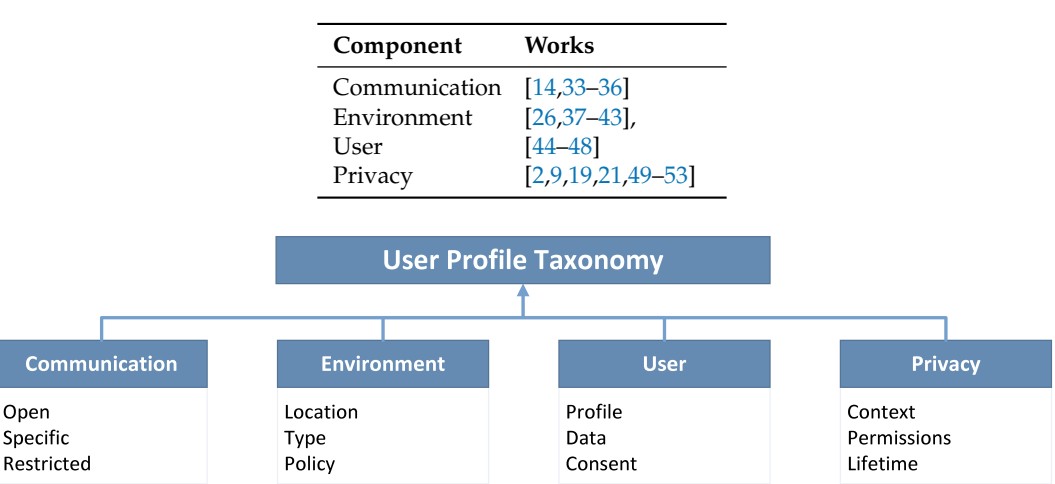

**Figure 2.** Taxonomic model for user profile.

### 4.1. Communication

The exchange of information between devices is essential in IoT applications and systems. IoT works with different forms of communication [33], allowing for smart environments to be always connected and providing a constant exchange of information. We classify the types of communication in:

- *Open Environments*: provide services that can be common to all users. The network technologies used for these environments must allow great mobility and minimal interference,

enabling multiple users and devices to be connected at the same time. Some protocols that cover these requirements include Wireless Networks (IEEE 802.11) and Mobile Data Network.

- *Specific Environments*: the identification of a specific environment requires precision. For this, protocols, such as Bluetooth, NFC (Near Field Communication), and RFID (Radio-Frequency IDentification), can be used.
- *Restricted Environment*: these environments require user identification and permission locally.

### 4.2. Environment

In an environment, there may be a variety of independent and cooperating devices to perform different tasks [37]. The environment that the user accesses, or is part of, is controlled and defines parameters and privilege criteria for user profiles. Thus, we classify the environment in:

- *Location*: reveals the geographical, physical, or virtual information of an environment [38].
- *Type*: used to classify the different types of environment [9].
- *Policies*: is a set of rules or conditions that define access and provide permission for a specific user in an environment.

### 4.3. Users

User information is increasingly necessary so that solutions can assist and fulfill their purpose properly. The existence of three classes is noticeable, namely:

- *Profile*: defines user behavior within ubiquitous environments [9].
- *Data*: includes is all user information.
- *Consent*: it is the means that an application has to provide the user with complete control of the collection, flow, and use of their personal data [47].

### 4.4. Privacy

Privacy has a great intrinsic value for its applicability in the real world. Given the great exchange of information in ubiquitous environments, the control and management of privacy are essential for modern applications [9,53]. We classify privacy as:

- *Context*: are the circumstances that accompany a fact or situation. In this case, the user is assigned permissions in an intelligent environment [21].
- *Permissions*: defines what information or actions the user allows to be accessed or transmitted from his profile.
- *Lifetime*: duration of the context in which a user is in a ubiquitous environment.

The proposed taxonomy assists in defining rules, policies, and parameters to provide the user profile in an intelligent environment. The application uses as a base the location where the user is, the environment's policies, the context the environment is in the user's permissions.

## 5. Architecure

The PRIPRO module (PRIvacy PROfiles) carries out transactions that progress and regress a given user profile. This module confronts common context identifiers, such as user name, environment identifier, and unique device identifier, with the rules established in the taxonomy. A remote procedure call communicates the result of this comparison to the other UbiPri modules.

By design, PRIPRO requires the same rigor that is applied to other elements of UbiPri, prioritizing lightweight communication and imposing acceptable criteria of integrity, control, and privacy to users and the environment.

The development of PRIPRO included the design and implementation of two different applications: the client (i.e., an Android app) and the web server. The former activates the controls of

the underlying operating system, allowing or blocking access to the device's resources according to the taxonomy in force in the environment. The latter receives the reading of the current state of the device under its control, compares it with the desired state for the device, and returns the set of necessary modifications in order to impose the appropriate context.

The sequence diagram of Figure 3 illustrates the integration of applications and the authentication request flow in the system. The client sends the user's metadata to the web server, which runs its `Identify()` method to processes the information received to decide on authentication. The web server also queries the user's consent, seeking this consent in the persistence layer or directly asking the user if it is their first interaction.

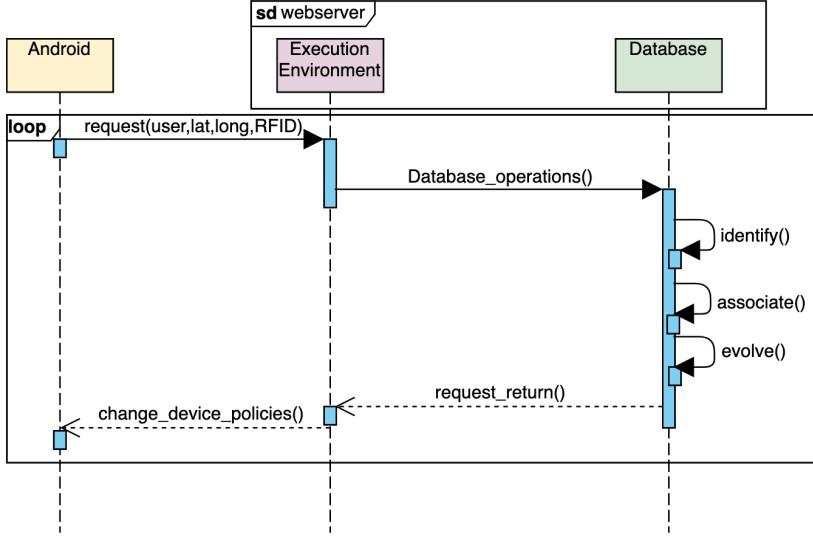

**Figure 3.** PRIvacy PROfiles (PRIPRO) system sequence diagram. The client app interacts with the web server for user profile management.

With consent, the web server runs the `Associate()` method to establish the context and policies of the environment and the permissions and lifetime of these definitions. Subsequently, the server runs the `Evolve()` method to evolve or regress the user's profile based on the environment's taxonomy. These adjustments remain stored in the web server database to enable the use of the settings in the environment and allow future consultations on evolution and regression history.

Figure 4 presents the state machine of the PRIPRO module. The first event is the user's authentication request. Subsequently, the module utilizes the user's location to determine in which environment he is. Finally, it applies the appropriate policy by directly combining the two characteristics of the relationship: user and location. The result is always a progression or regression of the profile.

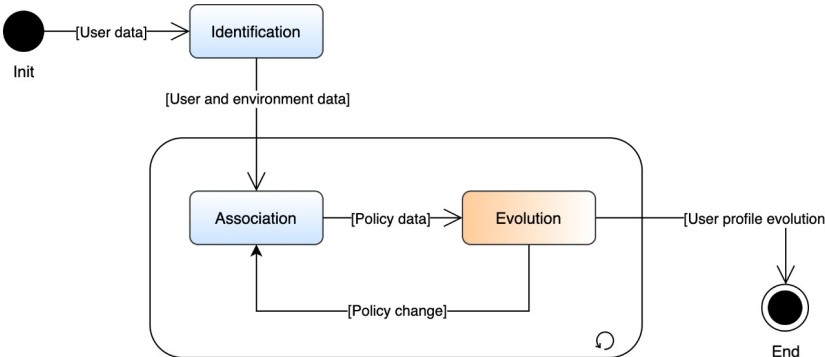

**Figure 4.** PRIPRO state machine. Each state represents the functionality of each sub-module of the web server.

*5.1. Client Application*

The client application comprises two sub-modules: Authenticator and Device Manager. The former authenticates the user with the web server, whereas the latter acts directly on the physical resources of the device, imposing the rules of the environment.

The logic of the Authenticator sub-module is comprehensive. It takes into account the geographic position taken from the device, the RFID identifier, and the pair of credentials (login and password) transmitted with mandatory encryption TLS (Transport Layer Security) 1.1 or later.

The Device Manager sub-module acts directly on the devices. The Android framework provides specific functionality that is very suitable for this purpose, also called device manager. An application that obtains permission to act as a device manager can act on any resource. These permissions enable the application to read from and write to the file system, operate or ban the camera, read geographic positioning, change Wi-Fi settings, generate calls, emit sounds, and everything else that the Android platform commonly provides in its application and system layers.

For implementing the client application, we utilized the Retrofit 2 library to generate HTTP requests and the GSON (Google JavaScript Object Notation Converte) library in order to convert Java objects into a neutral JSON (JavaScript Object Notation) representation suitable for transport.

Like other Android applications, the application's first interaction with the user presents a proposal for revoked permissions for its correct execution. The user must consent to the manipulations of PRIPRO on Android, so that the interaction occurs within the expected.

Both consent and user credentials are persisted in the Room Database area of the operating system. The approach consumes the operating system's safe, avoiding the future needs to grant privileges previously provided. Subsequent interactions occur directly in a task of the type Activity that provides an interface for the application.

The Google Location Services API provides a suitable mechanism for obtaining the geographic coordinates of the device and its user. The Android application's manifest file has been configured to contain the `ACCESS_COARSE_LOCATION` e `ACCESS_FINE_LOCATION` from the LocationManager. The result is a location determination that is based on the angulation of cell towers, Wi-Fi access points, and direct communication with satellites from the Global Positioning System (GPS), depending on the circumstances. Figure 5 presents the code snippet that collects the user's geographic coordinates in LAT/LONG format (Latitude and Longitude).

```kotlin
private suspend fun hasDeviceLocationChanged(
    lastUserLocation: Environment): Boolean {

    if (!isUsingDeviceLocation())
        return false

    val deviceLocation = getLastDeviceLocation()
        .await() ?: return false

    val comparisonThreshold = 0.003

    return Math.abs(
        deviceLocation.latitude - lastUserLocation.lat!!
    ) > comparisonThreshold && Math.abs(
        deviceLocation.longitude - lastUserLocation.lon!!
    ) > comparisonThreshold
}
```

**Figure 5.** Device location code.

The Device Manager API is essential to act on the device. It reaps the right to manipulate the characteristics of the device in favor of the application. Thus, it is inserted in a privileged level of consumption of resources of the operating system. Figure 6 presents a code fragment that changes the device's permissions.

```kotlin
class DeviceAdminReceiver : DeviceAdminReceiver() {

    companion object {
        private const val TAG = "DeviceAdmin"

        fun getComponentName(context: Context): ComponentName =
            ComponentName(context.applicationContext,
                DeviceAdminReceiver::class.java)
    }

    override fun onEnabled(context: Context, intent: Intent) {
        Log.i(TAG, "Enabled")
        Toast.makeText(context, "Device Admin: Enable",
            Toast.LENGTH_LONG)
        LoginActivity.launch(context)
        super.onEnabled(context, intent)
    }

    override fun onDisabled(context: Context, intent: Intent) {
        Log.i(TAG, "Disabled")
        Toast.makeText(context, "Device Admin: Disabled",
            Toast.LENGTH_LONG)
        super.onDisabled(context, intent)
    }
}
```

**Figure 6.** Device manager code.

*5.2. Web Server*

The web server comprises three sub-modules that implement the previously cited methods presented in Figure 3. The first sub-module is named Identifier and receives the requests for authentication from mobile devices and users. The second sub-module, called Associative, associates the privacy criteria that are defined in the environment taxonomies to users. Finally, the third sub-module, which is named Evolutionary, progresses or regresses the user's profile in the environment.

The web server receives and processes the HTTP protocol POST, GET, PUT, and DELETE calls that correspond to specific system routes. It implements actions of *create*, *read*, *update*, and *delete* for the tables *users*, *resources*, *environments*, and *policies*.

The Identifier sub-module routes the application. Each call sent to the server instance contains environment-related and user-related metadata. Subsequently, the Associative sub-module manages three key properties before deciding for a set of rules to apply: context, permission, and an expiry timestamp. Finally, the Evolutionary sub-module sends a response to the client containing the rules to be applied on the device.

We programmed the web server using Node.Js, a runtime environment that executes Javascript code outside a web browser for asynchronous and distributed applications. The event-oriented nature of Node.Js allows for operating the PRIPRO state machines directly in code and supporting a high number of data entry and exit operations. The PostgreSQL database provides persistence on this side of the application. This database has a native module, called PostGIS, to handle storage and processing geographic coordinates by direct set theory logic expression.

Algorithm 1 presents the inference for deciding which taxonomy to apply to a device based on the determination of its geographical position.

---

**Algorithm 1** Managging device and profile code

---

```
 1: for User logged do
 2:     Authorization (userId, lat, lon, rfid)
 3:     for Environment do
 4:         Check latitude and longitude
 5:         Calculate distance difference
 6:         Distance (Environment position, User position) <= Environment radius OR Environment RFId
    identification
 7:     end for
 8:     for Policy do
 9:         Verify active policy
10:         Verify active user profile
11:         if init_time <= TIME AND end_time >= TIME then
12:             Apply policy
13:         end if
14:     end for
15:     for User Profile do
16:         Check user profile active
17:         if userProfileActive ! = userProfileDefined then
18:             Evolves profile
19:         end if
20:     end for
21:     for Resource Permission do
22:         Check list of available resources
23:         if resource == allowed then
24:             Activate device resource
25:         end if
26:     end for
27: end for
```

---

If the user is within the border area of a taxonomy imposition, that taxonomy will be applied. From now on, a feature will only be available whether the environmental rules state that it is allowed.

## 6. Experimental Results

### 6.1. Materials and Methods

In the experiments, we used a laptop computer and three smartphones running different versions of Android OS (7.0, 7.1.1, and 8.1—the models available in the laboratory). The app can run on any version from 5.0 when the Android Device Administration API was introduced. We installed the client application on both devices with the necessary elevation of privileges to act as a system administrator.

We previously included user and taxonomy data in the database to carry out the operational tests. Besides, we chose the smartphone's rear camera to illustrate the state transition in which access to this feature should be allowed or blocked.

When a user enters a smart environment, the client application reads the geolocation. If it is the first time that the user enters the environment, the client application displays the consent dialog box to the user for entering authentication data. The client application then sends the collected data to the web server, which checks the current context and applies the environment policies, imposing control to the device's resources and performing the evolution of the user's profile.

We used different test scenarios in order to verify the correct functioning of the implemented codes and the integration between the client application and the web server. The following subsections present two of the scenarios used.

Finally, we utilized the Android Profiler tool of Android Studio 4.0 to profile the client application and obtain indicators regarding CPU and memory usage, data requirements, and energy consumption.

### 6.2. Scenario #1: Factory Guided Tour

The first scenario considered is a guided tour of an automotive factory. The factory receives visitors who must install the client application in their smartphones. They can visit three environments:

lobby, factory floor, and showroom, and cannot take photos or film the factory floor (the area where the ordinary workers work). Thus, we defined two contexts of smart environment: (*i*) *Normal*, for the lobby and the showroom; and (*ii*) *Restricted*, for the factory floor. We also defined two profiles for visitors: (*i*) *Basic*, which gives access to the rear camera, and (*ii*) *Guest*, which denies access to the camera. During the guided tour, the visitors' profile is updated to block or release their camera. This updating occurs according to the geolocation and the rules of each context. Figure 7 depicts how it works. When the visitor moves from a Normal to a Restricted context, PRIPRO changes its profile from Basic to Restricted Guest.

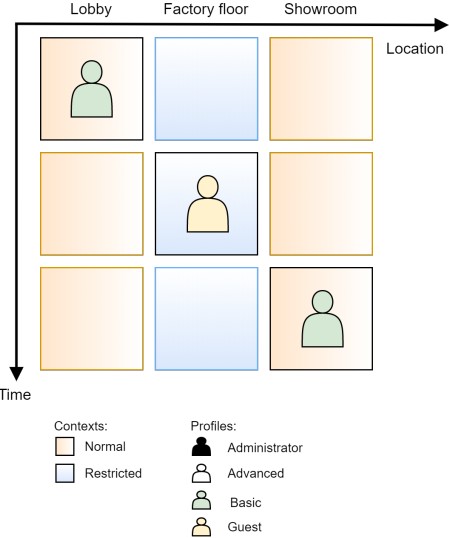

**Figure 7.** Scenario #1: Factory guided tour. The user's profile must be updated when he/she goes from one environment to another due to the change of context.

Figure 8 shows the result of an experiment validating the resulting permission on two devices. On Device 1, the user's profile is set to Basic, and the user has access to the rear camera. On Device 2, the user's profile is set to Guest, and the access to the camera is denied, as warned by the notification that is shown on the device's screen.

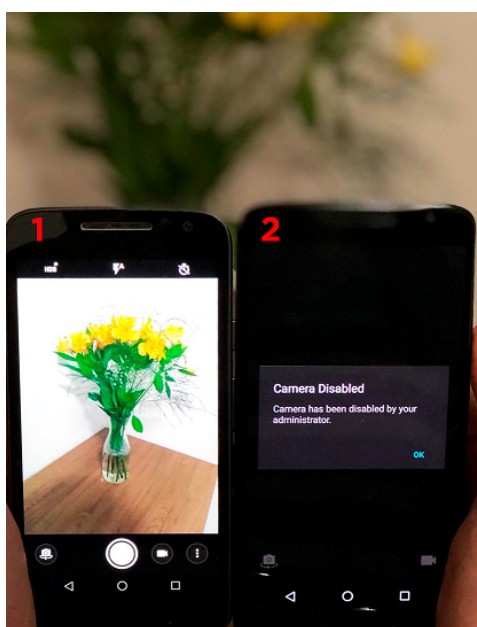

**Figure 8.** Digital camera access test. The camera is blocked according to the user's profile and the rules of the environment.

*6.3. Scenario #2: University Classroom*

The environment of the second scenario is a university classroom. In this scenario, the policies take into account the user type, geolocation, and the current time. According to the environment rules, students should not be able to use the rear camera of their smartphones to take pictures or film the classes. Subsequently, we defined two contexts: (*i*) *Restricted*, between the beginning and the end of classes; and (*ii*) *Normal*, for breaks between classes (as well as before or after class time). The profile of the lecturer is fixed to the Advanced level. On the other hand, the profile of the students can be set to Advanced or Basic. The latter denies access to the device's digital camera.

When entering the classroom, all of the students are authenticated by PRIPRO. At that moment, PRIPRO sets their profiles according to the current context. Before the beginning of the class, the context is Normal, and PRIPRO assigns the Advanced profile to all the students, freeing up full access to all the resources of their devices. However, at the beginning of a class, the environment context changes to Restricted. Thus, PRIPRO updates the students' profile to Basic and denies access to the device's rear camera. The students' profile returns to Advanced when the class ends. Figure 9 depicts how PRIPRO updates the students' profiles along the time. Next, Figure 10 illustrates the evolution of the user profile and the status of the resource access permission, showing that the rear camera is not available to the user when PRIPRO updates his/her profile to Basic.

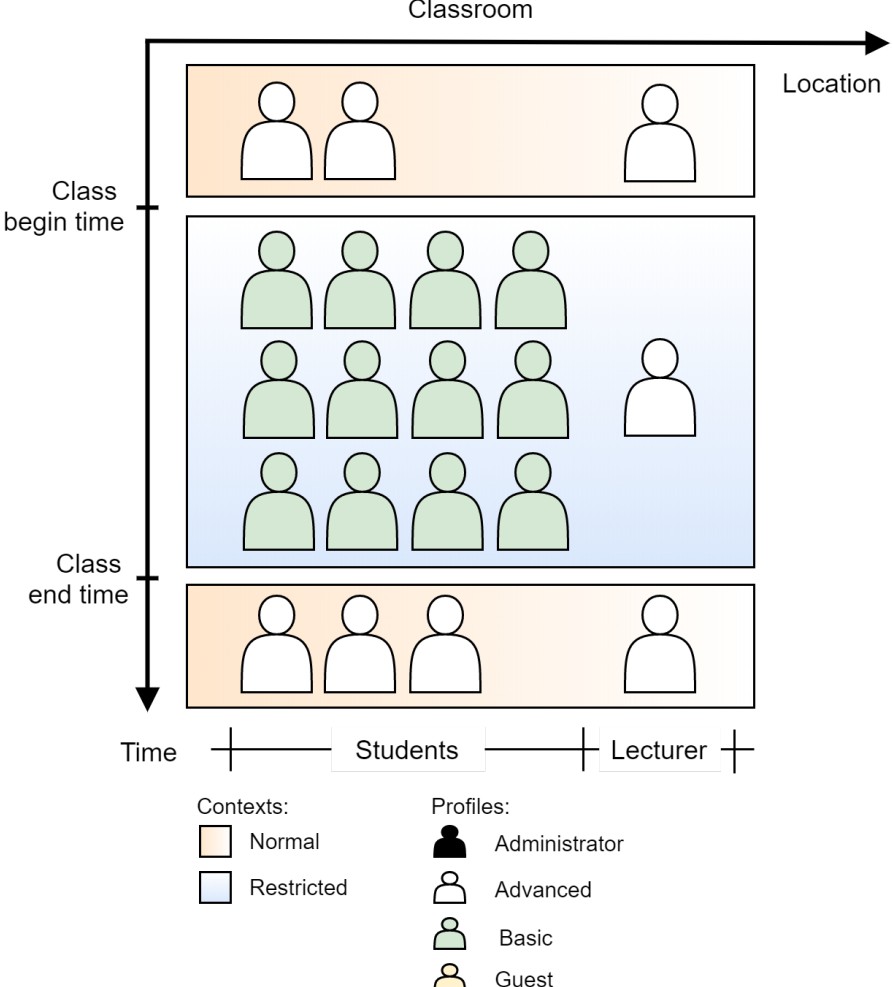

**Figure 9.** Scenario #2: University classroom. The Student user profile must be updated when the class begins due to the change of context.

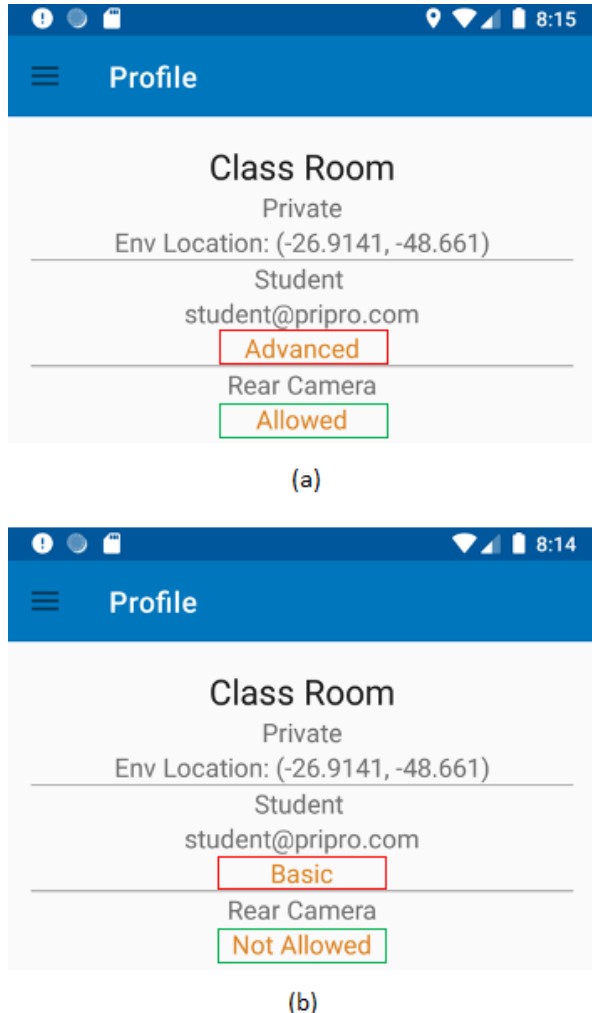

**Figure 10.** Test of scenario #2. The user's profile is updated from Advanced (**a**) to Basic (**b**) at the class begin time. Access to the rear camera is denied.

*6.4. Performance and Costs*

Figure 11 shows the profiling of the PRIPRO client application obtained using the Android Profiler tool. The execution of the app comprises three steps:

1.   Loading and connecting to the web server (from 0 to 5 ms).
2.   Processing and applying environmental rules to block the rear camera (from 5 to 13.5 ms).
3.   Standby operation (after 13.5 ms).

From Figure 11, we can see that the PRIPRO client app has a moderate CPU use (below 50%) and it reaches a peak memory footprint of around 373 MB (the reference smartphone has 2 GB of RAM). The amount of data transferred is very low, with data rates below 1.3 KB/s, and the energy consumption is light. These results indicate that the app has low response time and impact on the system operation, since each execution lasts less than 15 ms.

For comparison purposes, we profiled the GithubBrowserSample application available in the Samples section of the official Android Studio documentation. From the obtained results, as presented in Figure 12, we can see that the CPU usage and the energy impact are similar in both applications. However, the GithubBrowserSample has a lower peak memory footprint and a higher data transfer rate than the PRIPRO client app due to their different purposes.

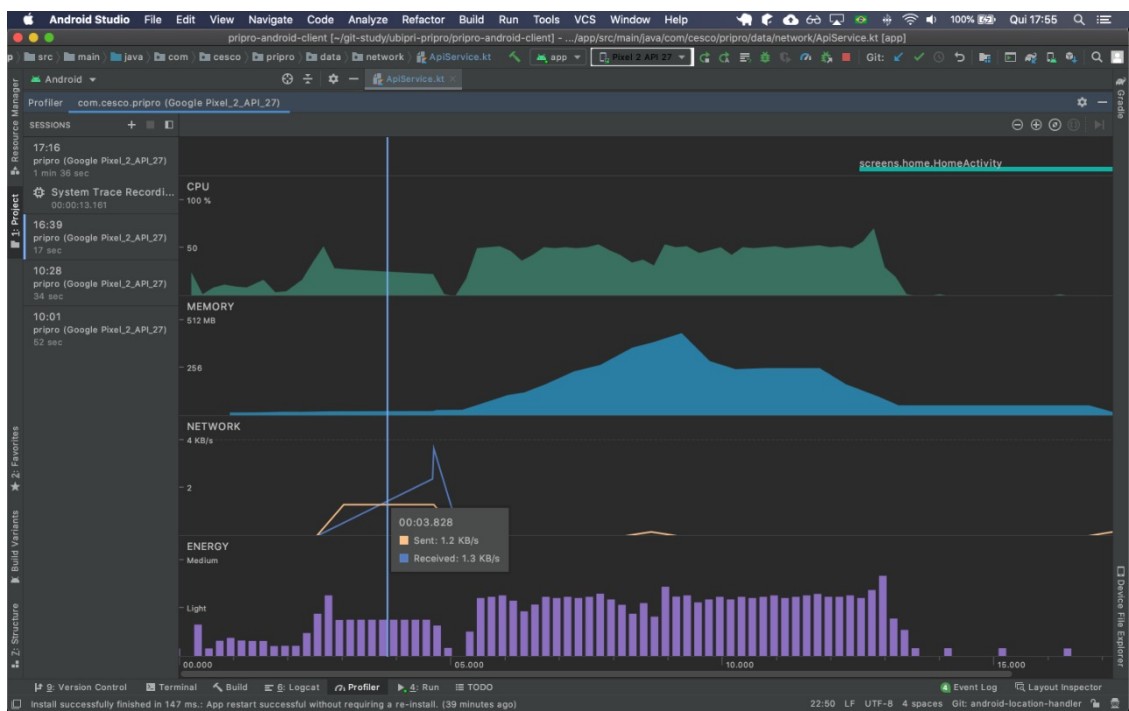

**Figure 11.** Profiling of the PRIPRO client application.

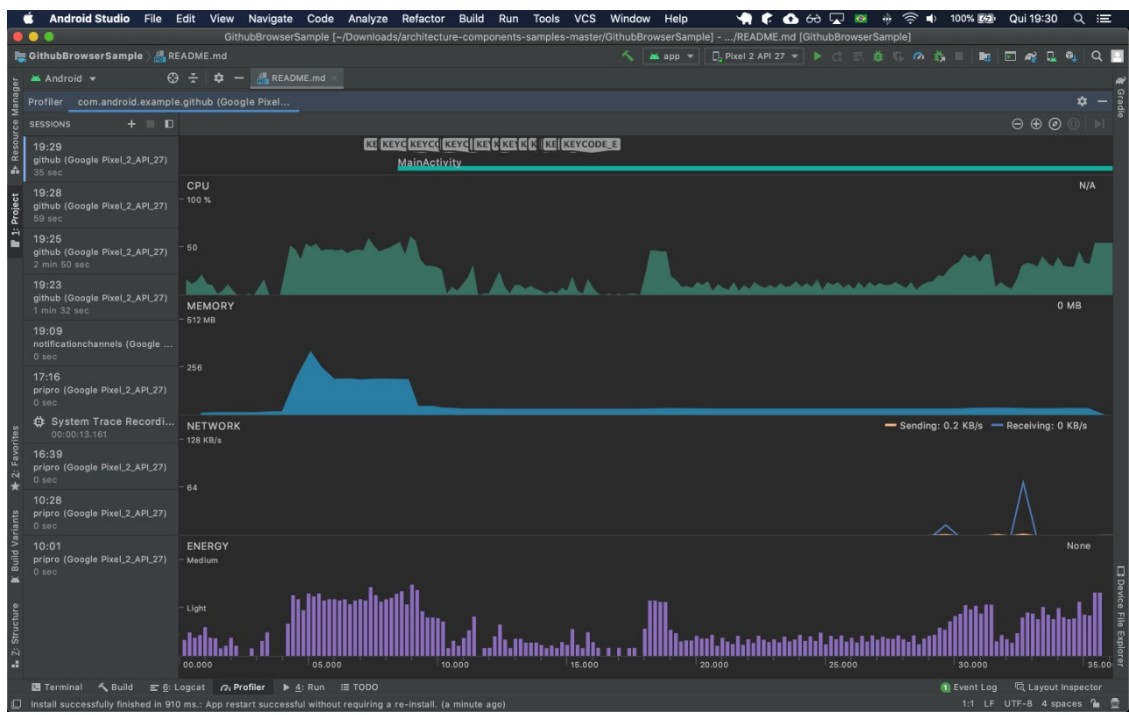

**Figure 12.** Profiling of the GithubBrowserSample application.

*6.5. Discussion*

Above, we presented two different scenarios used to evaluate the proposed implementation for PRIPRO. In the first scenario, we defined a static context for each environment, and the user's profile evolves as he/she changes from one environment to another. In the second scenario, we defined two contexts for the same environment, and the user's profile changes as time passes. These two examples illustrate the alternatives that PRIPRO offers to provide privacy in smart environments. In this sense, Table 3 summarizes a set of possible scenarios as well as the resources that could be automatically blocked by using the solution proposed, providing privacy, security, and comfort for users of smart environments.

**Table 3.** Possible denied resources for different scenarios.

| Environment | Cameras | Loudspeaker | Wi-Fi | Mobile Data | Calls | Emergency Call |
|---|:---:|:---:|:---:|:---:|:---:|:---:|
| Classroom | ● | ● | | | ● | |
| Museum | ● | ● | | | | |
| Airplane takeoff | ● | ● | ● | ● | ● | |
| Airplane in flight | | ● | | ● | ● | |
| City bus | | ● | | | | |
| Travel bus | | ● | | | | |
| Public library | ● | ● | | | ● | |
| Theater/Cinema | ● | ● | ● | ● | ● | |
| Hospital: Lobby | | ● | | | | |
| Hospital: ITU | ● | ● | ● | ● | ● | ● |
| Home: Dinner Room * | | ● | ● | ● | | |
| Restaurant * | | ● | ● | ● | | |

ITU: Intensive Therapy Unit. * With the context: "No WiFi! Talk to each other!".

## 7. Conclusions

Smart environments have made easier several human activities by providing smart services that automate everyday tasks. However, such environments require solutions for providing privacy and security to users and the environment itself. In this sense, this work presented the implementation of PRIPRO, a module of the UbiPri middleware that was designed for supporting the development of privacy mechanisms for smart environments.

The taxonomy presented in this article assists in defining the set of rules for the smart environment and updating the users' profile. Based on this taxonomy, we designed and implemented a solution that is composed of a client application to run on the user's smartphone and a web server to manage the evolution of the user's profile and permissions. The experiments demonstrated that the proposed solution provides a smart environment with the necessary service to manage the user's access to his/her smartphone's resources based on the rules of the environment. Such a solution can be used in several applications that require security and privacy for users and the environment. It should be noted that the presented taxonomy also provides coherent starting point for the definition of new environments and new user profiles, as well as event-oriented rules for progression and regression of the permissions associated with these profiles. Although this taxonomy and the implemented solution are inserted in the context of the UbiPri middleware, they can be applied to develop other middleware for smart environments with similar features.

As future work, we intend to carry out experiments that involve large amounts of data to evaluate the system's scalability. We also intend to evaluate the use of technologies that improve the user's geolocation in closed environments, such as 5G and UWB, to ensure the rapid recognition of his/her presence in a smart environment and apply the appropriate rules efficiently.

**Author Contributions:** Conceptualization, J.C., L.A.S., V.R.Q.L. and C.A.Z.; methodology, J.C. and V.R.Q.L.; validation, J.C., L.A.S. and F.B.; writing, original draft preparation, J.C., L.A.S., V.R.Q.L., and C.A.Z.; writing, review and editing, C.A.Z., F.B., M.N.-C. and V.R.Q.L. All authors have read and agreed to the published version of the manuscript.

**Funding:** The project 'Plataforma de Vehículos de Transporte de Materiales y Seguimiento Autónomo'—Target. 463AC03, SA063G19 Junta Castilla y León, Consejería de Educación and European Regional Development Fund (ERDF). This work was partially supported by Fundação para a Ciência e a Tecnologia under Project UIDB/04111/2020. Any opinions, findings, and conclusions or recommendations expressed in this material are those of the authors and do not necessarily reflect the views of the funding agencies.

**Acknowledgments:** Coordenação de Aperfeiçoamento de Pessoal de Nível Superior, Brasil (CAPES)—Finance Code 001, Conselho Nacional de Desenvolvimento Científico e Tecnológico (CNPq)—Grants 315287/2018-7 and 436982/2018-8, Fundação de Amparo à Pesquisa do Estado de Santa Catarina (FAPESC)—Grant 2019TR169. Laboratório de Telecomunicações de Portugal IT—Branch Covilhã.

**Conflicts of Interest:** The authors declare no conflict of interest.

## Abbreviations

The following abbreviations are used in this manuscript:

| | |
|---|---|
| ABAC | Attribute-Based Access Control |
| AC | Access Control |
| API | Application Programming Interface |
| ELM | Extreme Learning Machine |
| ESFCM | Semi-Supervised Fuzzy C-Means |
| GADAR | Group Activity Detection and Recognition |
| GPS | Global Positioning System |
| HTTP | Hyper Text Transfer Protocol |
| IoT | Internet of Things |
| IT | Information Technology |
| JSON | JavaScript Object Notation |
| NFC | Near Field Communication |
| NIST | National Institute of Standards and Technology |
| NGAC | Next Generation Access Control |
| OS | Operating System |
| PerBac | Pervasive-Based Access Control |
| PRICMU | Privacy Controll Management User |
| PRICOM | Privacy Communication |
| PRIDEV | Privacy Devices |
| PRIPRO | Privacy Profile |
| PRIADA | Privacy Adaptation |
| PRIENV | Privacy Environment |
| PRICRI | Privacy Criteria |
| PRIHIS | Privacy History |
| PRISEC | Privacy Security |
| PRISER | Privacy Service |
| JSON | JavaScript Object Notation |
| GNSS | Global Navigation Satellite System |
| GSON | Google JavaScript Object Notation Converter |
| OS | Operating System |
| RFID | Radio-Frequency Identification |
| SMD | Smart Mobile Devices |
| SSN | Spontaneous Social Network |
| UbiPri | Ubiquitous Privacy |
| UWB | Ultra Wide Band |
| WSN | Wireless Sensor Network |
| TLS | Transport Layer Security |

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
