# Peer review of "PRIPRO—Privacy Profiles: User Profiling Management for Smart Environments"

_electronics, doi:10.3390/electronics9091519_

Round 1
Reviewer 1 Report
In order to enhance the privacy of smart environments, the authors propose a privacy profiles approach and an android application on the smartphone by allowing or blocking resources according to the context. Experimental results show the proposed solution successfully communicates with the Android Device Administration framework, and the device appropriately reacts to the expected set of permissions. In my opinion, this work is meaningful and interesting for the privacy. However, how about the comparison with blockchain approach such as distributed blockchain-based trusted multidomain collaboration for mobile edge computing in 5G and beyond. I think the comparison between them in terms of centralized and decentralized solutions.
Author Response
Dear reviewer, we appreciate all your criticisms and suggestions that helped us to improve the quality of our research and, consequently, the article with the results obtained. Attached I send the answers to your questions and requests, together with the changes requested by the other reviewers. With that, we believe that the article is now able to be published.
* see the attached file please have all the answers and comments.

Reviewer 2 Report
I recommend this article for minor revision due to following cause
- Background missing. Clarity of presentation is needs to improve via clearly specifying objectives, focus and motivation of the paper.
- The abstract of the paper needs to be rewrite.
- It would be better to performance comparison with recent studies and recent techniques in the context of the proposed work.
- The overall workflow of the paper should be improved.
- The author should also refer/cite the following recent works in the Smart Environment.
- Xie, K., Ning, X., Wang, X., He, S., Ning, Z., Liu, X., ... & Qin, Z. (2017). An efficient privacy-preserving compressive data gathering scheme in WSNs. Information Sciences, 390, 82-94
- Zeng, W., Chen, P., Chen, H., & He, S. (2016). PAPG: Private Aggregation Scheme based on Privacy-preserving Gene in Wireless Sensor Networks. KSII Transactions on Internet & Information Systems, 10(9).
- https://www.sciencedirect.com/science/article/pii/S1568494618303508 (Semi-supervised learning based distributed attack detection framework for IoT).
- https://www.sciencedirect.com/science/article/pii/S0020025517309106 (Social network security: Issues, challenges, threats, and solutions
Author Response
Dear reviewer, we appreciate all your criticisms and suggestions that helped us to improve the quality of our research and, consequently, the article with the results obtained. Attached I send the answers to your questions and requests, together with the changes requested by the other reviewers. With that, we believe the article is now able to be published.
* see the attached file please have all the answers and comments.

Reviewer 3 Report
Line 1-2 – Abstract – the first sentence is a little odd and needs rewording
Overall – the paper is well written. The content is easy to follow and understand. The literature review is pertinent to the theme of the paper. The results are well presented.
Have you considered power consumption on the phones when using the app?
Author Response

(The authors gave the same response as above.)

Reviewer 4 Report
The paper “PRIPRO – PRIVACY PROFILES User Profiling Management for Smart Environments” presents a user profiling management system.
Even if the paper is interesting, in my opinion the paper has many gaps and needs to be restructured. The introduction of the paper is very long and confuses the reader. If this was the first approach presented in the scientific literature the contribution would be clear
Many sentences are poor constructed: “According to [15], the number of smartphone users worldwide equals 3.5 billion in 2020.”. We are in the last half of the year 2020 so is not a prediction.
Many acronyms are not explained in the text.
There are a lot of data that is not clearly presented and confusing.
The last part of the paper where the design of the profile management is superficial written.
Although the addressed research area is of interest, the paper is not clear on the innovation it brings. This might be due to a poor organization and description of the developed work.
The authors must restate the main contribution of the paper.
I believe the paper is not ready for publication.
Author Response

(The authors gave the same response as above.)

Reviewer 5 Report
The paper is well written with interesting contribution and marginal novelty.
The introduction can be improved by combining it with the related works section. more recent works can be also added to review the limitations of the existing works and highlight motivation behind your proposed contribution.
Author Response

(The authors gave the same response as above.)

Round 2
Reviewer 1 Report
The paper can be accepted in this version.
Reviewer 4 Report
I read the paper.
The revision is performed correct.
My final decision in to accept paper.